# An open source tool to infer epidemiological and immunological dynamics from serological data: serosolver

James A. Hay [1,2]*, Amanda Minter [3], Kylie E. C. Ainslie [1], Justin Lessler [4], Bingyi Yang [5,6], Derek A. T. Cummings [5,6], Adam J. Kucharski [3☯], Steven Riley [1☯]*

**1** MRC Centre for Global Infectious Disease Analysis, Department of Infectious Disease Epidemiology, School of Public Health, Imperial College London, London, United Kingdom, **2** Center for Communicable Disease Dynamics, Department of Epidemiology, Harvard T. H. Chan School of Public Health, Boston, Massachusetts, United States of America, **3** Centre for the Mathematical Modelling of Infectious Diseases, London School of Hygiene & Tropical Medicine, London, United Kingdom, **4** Department of Epidemiology, Johns Hopkins Bloomberg School of Public Health, Baltimore, Maryland, United States of America, **5** Department of Biology, University of Florida, Gainesville, Florida, United States of America, **6** Emerging Pathogens Institute, University of Florida, Gainesville, Florida, United States of America

☯ These authors contributed equally to this work.
\* jhay@hsph.harvard.edu (JAH); s.riley@imperial.ac.uk (SR)

**Data Availability Statement:** The serosolver R package is available at: https://seroanalytics.github.io/serosolver/. All data used in these analyses are available within the R package git repository at:

## Abstract

We present a flexible, open source R package designed to obtain biological and epidemiological insights from serological datasets. Characterising past exposures for multi-strain pathogens poses a specific statistical challenge: observed antibody responses measured in serological assays depend on multiple unobserved prior infections that produce cross-reactive antibody responses. We provide a general modelling framework to jointly infer infection histories and describe immune responses generated by these infections using antibody titres against current and historical strains. We do this by linking latent infection dynamics with a mechanistic model of antibody kinetics that generates expected antibody titres over time. Our aim is to provide a flexible package to identify infection histories that can be applied to a range of pathogens. We present two case studies to illustrate how our model can infer key immunological parameters, such as antibody titre boosting, waning and cross-reaction, as well as latent epidemiological processes such as attack rates and age-stratified infection risk.

## Author summary

Antibody levels can determine previous exposure to a pathogen and how likely individuals are to be infected in the future. However, antibody concentrations change over time, and some pathogens are continually evolving. In such cases, individuals may be infected and vaccinated multiple times when their pre-existing immunity fails, leading to a wide range of antibody profiles. Traditional approaches to analyse such data do not typically account for this. In addition, studies collecting antibody data may be designed differently, but are often underpinned by similar biological processes. We developed a statistical method and

https://github.com/seroanalytics/serosolver/tree/master/inst/extdata. All analyses and code used in the main text are included in the accompanying Supporting Information case studies.

**Funding:** For funding we acknowledge: UK Medical Research Council (MRC) and the UK Department for International Development (DFID) under the MRC/DFID Concordat agreement, also part of the EDCTP2 programme supported by the European Union (UK, Centre MR/R015600/1) (JAH, KECA and SR); Wellcome Trust Investigator Award (UK, 200861/Z/16/Z) (SR); Wellcome Trust Collaborator Award (UK, 200187/Z/15/Z) (SR); National Institute for General Medical Sciences (US, MIDAS U01 GM110721-01) (SR); National Institute for Health Research (UK, for Health Protection Research Unit funding) (SR); Sir Henry Dale Fellowship jointly funded by the Wellcome Trust and the Royal Society (grant Number 206250/Z/17/Z) (AJK); DATC acknowledges support from NIH (R56AG048075); DATC and BY acknowledge support from NIH (R01AI114703); JL acknowledges support from NIH National Institute of Aging (R56, AG048075-01A1). The funders had no role in study design, data collection and analysis, decision to publish, or preparation of the manuscript.

**Competing interests:** The authors have declared that no competing interests exist.

accompanying software package to better understand the immunology and epidemiology of these complex systems using serological data. We present two case studies to demonstrate how our software package, *serosolver*, can be applied to different settings: i) the epidemiology of the 2009 pandemic A/H1N1 influenza virus in Hong Kong and ii) historical patterns of A/H3N2 influenza infection in Guangzhou, China. These results demonstrate how modern analytical methods can reveal additional information from serological data that is otherwise missed using traditional approaches.

## Introduction

Serological assays measure the interaction of a virus with the antibody repertoire of an individual host [1]. Originally developed in the mid-20th century, assays based on haemagglutination inhibition (HI) and viral neutralization (NT) are still widely used and demonstrate good intra-laboratory reproducibility [2, 3]. These assays can be setup relatively easily once viral stocks are in place, allowing antibody concentrations to be quantified quickly and inexpensively [4]. Usually, sera are diluted in successive 2-fold steps and mixed with a fixed amount of virus [4]. Inhibition of viral activity at higher serum titres indicates a strong antibody response, whereas failure to inhibit activity at the lowest titre indicates the absence of a significant response. The longevity of antibodies make serological assays a key tool in epidemiological surveillance [5–8].

There are two common ways of interpreting antibody titres in serosurveillance: threshold metrics and titre rises. When only a single sample is available for an individual, a threshold titre for 'seropositivity' is often used as evidence of prior exposure or protection or both, for example the commonly used HI titre threshold of 1:40 for influenza [4, 9, 10]. When serum samples encompassing a window of known strain circulation are available, a $\geq$ 4-fold rise in antibody titre is usually interpreted as exposure to that strain [4]. Samples taken before and after an influenza season for which the main circulating strain is known can therefore be used to infer attack rates [11–13]. Given that there is a degree of subjectivity when interpreting the serum dilution series, a $\geq$ 4-fold difference, within a 2-fold dilution scheme, is deemed to be more robust against human error than a $\geq$ 2-fold difference in assessing the presence of haemaglutination (for HI) or cell death (for NT) in each well of the assay plate [14, 15]. However, a Bayesian analysis of titre rise data suggested that the somewhat arbitrary 4-fold rise misses a substantial number of infections that result in lesser titre rises [16]. Individual-level differences in age, infection history, time between exposure and measurement, and virus-specific effects likely all play a role in generating sub-4-fold titre rises [17–19].

Cross-reactivity complicates the interpretation of serological results when an individual may have been exposed to two or more antigenically related viruses. Two pathogens are considered antigenically related if exposure to one generates a cross-reactive antibody response to the other in a serological assay. For example, antibodies generated in response to infection with one dengue virus serotype can cross-react to viruses of another serotype [20], as well as other flaviviruses such as Zika virus [21, 22]. Moreover, lineages of successive circulating influenza A strains cross-react with their precursors and progeny of the same subtype [23].

Interpretation of data from panels of cross-reacting strains has improved through antigenic cartography: a method to reduce complex tables of HI readings for novel viruses and reference antisera to two dimensional space visualised as an 'antigenic map' [20, 24, 25]. An individual's entire antibody repertoire against an antigenically variable pathogen can then be projected as a surface over these antigenic maps, with the height of the surface indicating the expected titre for that individual against a strain at any location in the map [26]. These 'antibody landscapes'

can be used to generate biological insight by investigating how antibody profiles develop over an individual's life [27].

Recently, there have been a number of initiatives to refine analyses of serological data. Mathematical models that harness the qualitative predictability of the post-exposure antibody response (boosting and subsequent waning of antibody levels) retain much of the information inherent in antibody titres that is otherwise lost using threshold or 4-fold rise metrics, enabling improved inference of unobserved single infections [28]. These methods have been applied in a number of human and wildlife disease systems to understand antibody waning rates and population trends in infection [13, 29–33]. However, dynamical models for antigenically variable pathogen systems with the potential for multiple exposures and cross-reactive antibody responses have been somewhat neglected until recently [27, 34–36].

Here, we present the R package *serosolver*, which is the latest version of a code base developed to increase the epidemiological insight available from serological assays [27, 37]. The *serosolver* package takes assay results from one or more serum samples for an individual, which may have been tested against one or more related viral strains, and infers a history of infections for that individual that is consistent with the observed titres. It can jointly estimate the parameters for the antibody kinetics model by simultaneously inferring infection histories for many people. Our approach introduces a number of refinements over existing methods designed to support modelling of multi-season, antigenically variable pathogen systems, including a well-defined statistical framework to represent multiple exposures and the inclusion of cross-reactive antibody responses. We use a Bayesian approach and obtain samples from the joint posterior distribution of infection histories and antibody kinetics parameters. The required assumptions for some priors are straightforward and may incorporate previously observed immunological phenomena. Prior assumptions for infection histories and the process that generates them can also be incorporated, but require additional justification, as we shall discuss.

First, we outline how the joint posterior distributions for antibody kinetics parameters, individual infection histories and the time-varying probability of infection in the population are flexibly implemented in the *serosolver* package. We then show how the package can be applied to cross-sectional and longitudinal influenza data from mainland China and Hong Kong to infer key epidemiological and immunological values.

## Methods

### Approach

The methods underpinning *serosolver* are motivated by the following base assumptions: (i) antibody titres may be measured using serum samples taken at some point in time; (ii) these antibody titres are an incomplete observation of true, underlying antibody levels that undergo a dynamical process following infection; (iii) these underlying antibody kinetics arise from the culmination of repeated exposures to antigenically related or identical pathogens. The aim is infer the combination of infections at different times or with different strains that are most consistent with observed antibody titres. In the main text, we describe how this system is implemented specifically for the subsequent case studies on influenza. S1 Text describes the framework in a more generalised form as a reference for future development of *serosolver* to other disease systems.

We frame the overall inference challenge as obtaining estimates for the joint posterior distribution of antibody kinetics parameters ($\Theta$), individual infection histories ($Z$) for all $n$ individuals in the sample, and the time-varying probability of infection in the population ($\Phi$) across $m$ possible discrete infection periods given an observed serological dataset ($Y$, which

may include assay measurements against one or more strains). This distribution, $P(Z, \Phi, \Theta|Y)$, is comprised of three components:

1. The combined observation and antibody kinetics models $f(Y_i|Z_i, \Theta)$, which give the likelihood of observing a set of titres $Y_i$ for each individual $i$ at discrete serum sampling times ($t_i$) given infection history $Z_i$ and the antibody kinetics parameters $\Theta$;

2. The infection history model $P(Z_{i,j}|\Phi_j)$, which gives the probability of individual $i$ having been infected with strains circulating in each discrete time period $j$ when infection might have occurred (between time $j_{min}$ and $j_{max}$), conditional on the time-varying population infection probabilities $\Phi$;

3. The prior probabilities of the antibody kinetics parameters, $P(\Theta)$, and the prior probability of any infection in each discrete time period $j$, $P(\Phi_j)$.

$$P(Z, \Phi, \Theta|Y) \propto \prod_{i=1}^{n} \Big( \overbrace{\prod_{t \in t_i} \underbrace{f(Y_{i,t}|Z_i, \Theta)}_{\substack{\text{(i) Observation} \\ \text{and antibody kinetics} \\ \text{models}}}}^{\substack{\text{Serum samples} \\ \text{taken at a subset} \\ \text{of all time periods}} \overbrace{\prod_{j=j_{min}}^{j_{max}} \underbrace{P(Z_{i,j}|\Phi_j)}_{\substack{\text{(ii) Infection} \\ \text{history model}}}}^{\substack{\text{Infection history for all} \\ \text{possible infection times} \\ \text{or strains}} \underbrace{P(\Phi_j)}_{\substack{\text{(iii) Priors on infection} \\ \text{probability and} \\ \text{antibody kinetics} \\ \text{parameters}}} \Big) P(\Theta) \tag{1}$$

In each discrete time period, $j$, we assume that there is only one strain that circulates. Reference to $j$ therefore refers to both the time period itself and the index of the strain that circulated during that time. Treating time as discrete differs to some previous approaches which model infection times as continuous variables [13, 35]. The time resolution of $j$ can be set when running *serosolver* depending on the amount of data; using only one possible infection period ($m = 1$) is conceptually similar to an analysis of seropositivity, whereas choosing $m$ to represent many small intervals of time (e.g. months) becomes conceptually similar to continuous time.

## Antibody kinetics model

For a given individual infection history and set of biological parameters, the antibody kinetics model generates a set of expected log titres for that individual against all possible test strains. These antibody titres are observed at only a subset of times for which serum samples are available, and the model-predicted antibody titres across all times are therefore referred to as latent antibody titres. Although other functions for $f(Y_{i,t}|Z_i, \Theta)$ may be implemented and used with only minor modifications to the code, the model used here follows previous work [27]. The expected log titre individual $i$ has against the strain that circulated during discrete time period $j$ when observed at time $t$ ($X_{i,j,t}$) is defined as a linear, deterministic combination of contributing antibody responses from each prior infection:

$$X_{i,j,t} = \sum_{k \in Z_i} Z_{i,k} \ s(Z_i, k)[\mu_l d_l(j, k) + \mu_s w(t, k) \ d_s(j, k)] \tag{2}$$

The model components are defined by:

1. Long-term boosting defined by a parameter $\mu_l$, giving the expected persistent rise in titre against a homologous strain following primary infection.

2. Short-term boosting. The transient component of the antibody kinetics model defined by $\mu_s w(t, k) = \mu_s \max\{0, 1 - \omega(t - k)\}$, where $\mu_s$ is the boost in homologous titre, $\omega$ is a waning parameter to be fitted, and $t - k$ is the time since infection with strain $k$.

3. Cross-reactive antibody responses from related strains. We assume the level of cross-reaction between a test strain $j$ and infecting strain $k \in Z_i$ decreases linearly with antigenic distance (defined below) [24]. The cross-reaction functions are $d_l(j, k) = \max\{0, 1 - \sigma_l \delta_{j,k}\}$ and $d_s(j, k) = \max\{0, 1 - \sigma_s \delta_{j,k}\}$ for the long-term and the short-term boosts respectively. $\delta_{j,k}$ is the antigenic distance between strains $j$ and $k$, and $\sigma_l$ and $\sigma_s$ are fitted parameters.

4. Antigenic seniority from boosting suppression. This results in lower titre boosts from later compared to earlier infections. In the model, the contribution of an exposure is scaled by $s (Z_i, k) = \max\{0, 1 - \tau(N_k - 1)\}$, where $N_k$ is the infection number (i.e. primary infection is 1, secondary is 2) and $\tau$ is a fitted parameter.

In the model, the antigenic distance $\delta_{j,k}$ between strains $j$ and $k$ is defined by a matrix of pairwise distances in an antigenic map. $\delta_{j,k} = 1$ if serum from a naive ferret infected with strain $k$ gives a titre against strain $j$ one log unit lower than against the homologous strain $k$ [24]. The *serosolver* model can accommodate antigenically varying strains (all $\delta_{j,k}$ are specified) or a single homologous strain (all $\delta_{j,k} = 0$). The extent to which strains are antigenically distinct or similar can be specified by the distance matrix.

The antibody kinetics model can be reduced to simpler models by setting certain parameter values equal to 0. These nested sub-models allow hypothesis testing to distinguish between immunological mechanisms. For example, a model without antigenic seniority can be created by setting $\tau = 0$ or a model with only waning responses by setting $\mu_l = 0$. In addition, *serosolver* can be extended to include more complex antibody kinetics (e.g. boosting that scales as a function of pre-existing titre), as described in S2 Text. We note that the additional immunological phenomena described in S2 Text are not exhaustive, and additional mechanisms may be easily implemented by making minor modifications to the package code.

## Observation model

The expected titre $X_{i,j,t}$ defined in Eq 2 feeds into the observation model. For HI and NT titres, this necessitates conversion of continuous latent titres into discrete observations. The distribution of the observed titre consists of a normally distributed random variable $g(s)$ with mean $X_{i,j,t}$ and variance $\varepsilon$, which is then censored to account for integer-valued log titres in the assay. Hence the probability of observing an empirical titre at time $t$ within the limits of a particular assay $Y_{i,j,t} \in \{0, \ldots, Y_{\max}\}$ given expected titre $X_{i,j,t}$ for individual $i$ measured against strain $j$ at serum sampling time $t$ is:

$$P(Y_{i,j,t}|X_{i,j,t}) = f(Y_{i,j,t}|Z_i, \theta) = \begin{cases} \int_{Y_{i,j,t}}^{Y_{i,j,t}+1} g(s)ds & \text{if} \quad Y_{i,j,t} \in \{1, Y_{\max} - 1\} \; ; \\ \int_{-\infty}^{1} g(s)ds & \text{if} \quad Y_{i,j,t} = 0 \; ; \\ \int_{Y_{\max}}^{\infty} g(s)ds & \text{if} \quad Y_{i,j,t} = Y_{\max} \; . \end{cases} \tag{3}$$

There are additional options in *serosolver* to include strain-specific measurement bias, which may arise through strain-specific differences in assay reactivity [26, 38, 39]. Specifically, an additional observation error is added to the predicted log antibody titres; this measurement error can be different for each individual strain or can be specified for a group (or cluster) of

strains. The predicted titre $X'_{i,j,t}$ taking into account strain-specific measurement bias is:

$$X'_{i,j,t} = X_{i,j,t} + P_j \tag{4}$$

where $P_j$ is the measurement offset for strain $j$. $P_j$ may be estimated as independent parameters for each $j$, or may be assumed to come from a hierarchical distribution $\rho_j \sim \mathcal{N}(\bar{\rho}, \sigma_\xi^2)$. $P_j$ may also be fixed for particular strains/groups e.g. fixing $\bar{\rho} = 0$ or $\rho_{j_{max}} = 0$.

## Infection history model

Each individual's infection history is tracked by *serosolver* as a vector of binary latent states indicating the presence (1) or absence (0) of infection within each unit of discrete time. $Z_i = [Z_{i,1}, Z_{i,2}, \ldots, Z_{i,j \leq t}]$ represents a vector of infection states that could have occurred at or before the time a serum sample is taken during interval time $t$. The set of infection histories for the sample population is therefore described by a binary matrix, $Z = [Z_1, Z_2, \ldots, Z_n]$. Each row of the matrix represents an individual, $i$, and each column represents a time, $j$, at which an individual could be infected once. The likelihood for the infection history model, $P(Z|\Phi)P(\Phi)$, is given by:

$$P(Z|\Phi)P(\Phi) = \prod_{i=1}^{n} \prod_{j=j_{min}}^{j_{max}} P(Z_{i,j}|\Phi_j)P(\Phi_j) \tag{5}$$

where each infection event, $Z_{i,j}$, is the outcome of a single Bernoulli trial with probability $P(Z_{i,j}|\Phi_j = \phi_j) = \phi_j^{Z_{i,j}}(1-\phi_j)^{(1-Z_{i,j})}$. A value of $z_{i,j} = 1$ indicates that individual $i$ was infected in discrete time period $j$ and $z_{i,j} = 0$ indicates that they were not. The choice of the prior distribution for the probability of infection, $P(\Phi_j)$, is discussed below and in further detail in S1 Text. The time resolution of infection times may be set by the user depending on the data: frequent serum sampling times affords greater time resolutions (e.g. months), whereas less frequent sampling may be better suited to cruder time resolutions (e.g. years).

The infection history posterior can be used to calculate the population attack rate over time. Attack rates can be inferred through combining estimated infection histories post-hoc to estimate the proportion of at risk individuals that were infected in a given time period. Summing the columns of the infection history matrix gives the total number of infections for a given time period, whereas summing the rows give the total number of lifetime infections for an individual. To ensure biological plausibility, individual infection histories are constrained to prevent infections before an individual is born and after the last time at which a serum sample was taken. A key feature of the package is that the user is given control over the prior assumptions for the infection history and the probability of infection during each time period (months, years etc).

## User inputs

**Data.** The *serosolver* package requires a dataset of individual-level log titres. Each individual may have repeat titre measurements from one or more serum sampling times, $t$ (i.e. when each serum sample was collected), against viruses that may have circulated at each discrete time period $j$ (i.e. when the strain was originally isolated).

The initial development of *serosolver* focused on influenza A/H3N2, which has circulated in human populations since 1968 and has undergone substantial antigenic evolution over this time [24, 39–41]. Fig 1 illustrates how our analytical approach applies to influenza A/H3N2. In *serosolver*, a key but non-essential input to the framework is therefore an antigenic map: a data

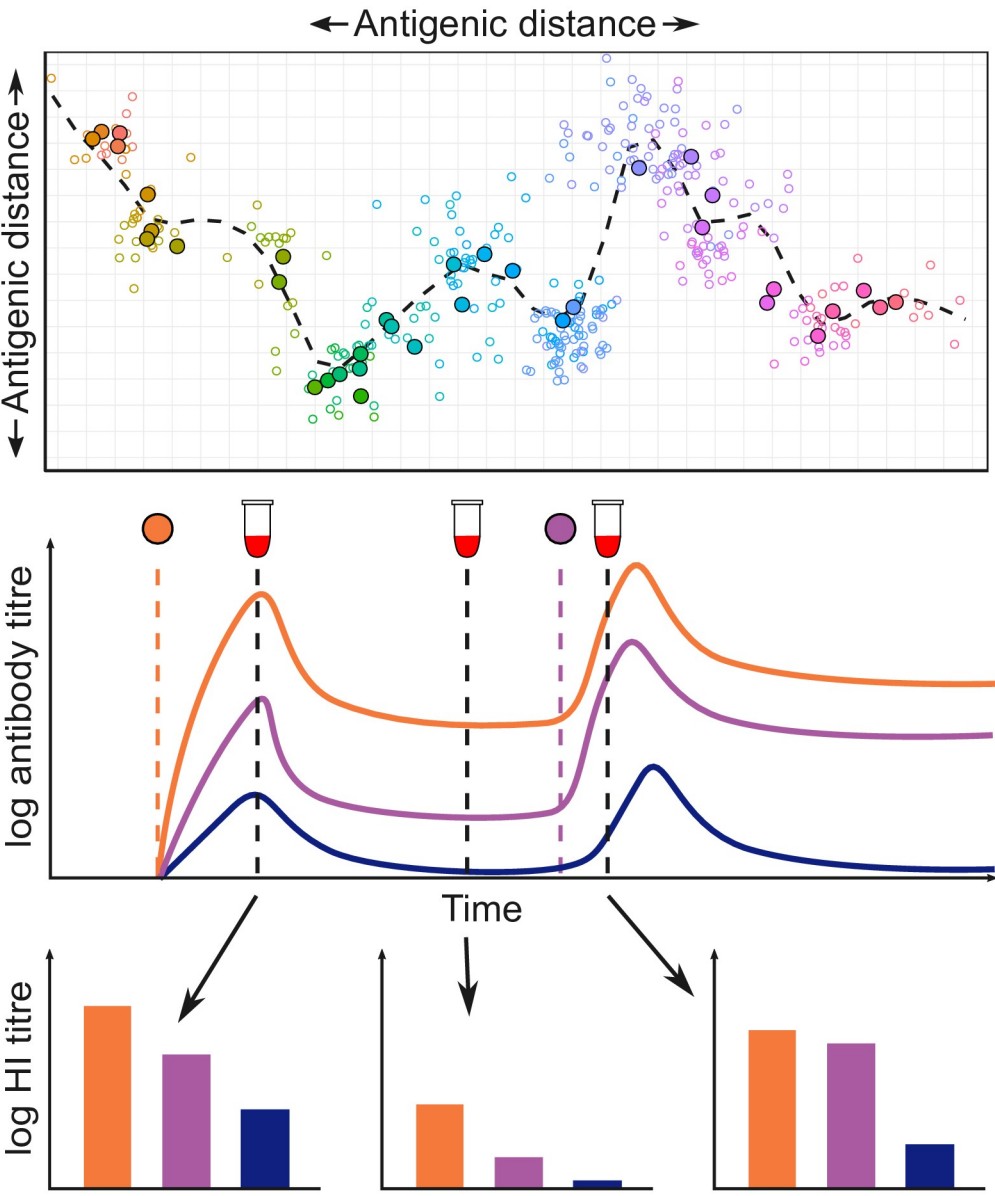

**Fig 1. Conceptual overview of the analytical approach used in *serosolver*, as applied to influenza A/H3N2. Top panel**: antigenic map for influenza A/H3N2 using coordinates from [24], with different viruses coloured by year of isolation. Solid points show centroids across all strains isolated in a given calendar year, hollow points show individual strains. Dashed line shows an antigenic summary path, generated by fitting a smoothing spline through the observed isolates. Points further apart in space are less cross-reactive. **Middle panel**: conceptual illustration of the antibody kinetics model. An individual is infected with the orange virus, which results in boosting and waning of homologous antibody titres. In parallel, antibodies that cross react with viruses at different points in antigenic space also boost and wane (purple and blue viruses). The individual is later infected by the purple virus, which leads to further boosting and waning of antibodies. **Bottom panel**: HI titres measured from serum samples taken at different times capture different parts of the homologous and cross reactive antibody kinetics. Different sampling strategies will represent different subsets of these measurements e.g. a cross-sectional study might inform a single subplot, whereas a longitudinal study might inform just the orange bars from each of the three subplots. Clearly a sampling strategy with multiple serum samples and many viruses tested per sample will provide the most information.

structure describing the two-dimensional location of viruses in antigenic space that circulated at each time point during the period of interest (specifically, an $x$ and $y$ coordinate for the strain that circulated in each time period $j$). This map is used to calculate the pairwise antigenic distance between any two viruses (i.e. $\delta_{j,k}$ in the antibody kinetics model, for strains $j$ and $k$). Not all disease systems or epidemiological problems require an antigenic map. For example, the map may be considered equivalent to a single point when the pathogen of interest shows no antigenic variability over time. A model with only one circulating antigenic variant may therefore be specified by using a null input for the antigenic map argument.

**Prior assumptions.**   We take a Bayesian approach in *serosolver*, meaning that priors must be defined for all model parameters and infection histories. The priors on the antibody kinetics parameters are uniform by default, but users may create their own prior function, which may be based on previous analyses. For example, constrained estimates for the short-term antibody waning parameters may be used to specify strong beta or Gaussian priors on antibody kinetics parameters.

Priors on the infection histories require more consideration, because the prior also captures any assumptions regarding the infection generating process. Because the number of potential infection times and strains can be vast, the contribution of the infection history prior must be well characterised to avoid any unforeseen bias during inference. The prior assumption on the functional form of $\boldsymbol{\Phi}$, whether individual infection risks are independent at a given time $j$, and whether an individual's risk of infection depends on infection outcomes at previous times can have important implications for the prior on key infection history summary metrics, such as the attack rate in a given time period and the lifetime number of infections for an individual.

There are four infection history prior options in *serosolver*. We summarise these priors here and discuss their trade-offs in the Results section through simulation-recovery experiments, though an extensive discussion of their derivation is provided in S1 Text.

**Prior 1, hyper-prior on the probability of infection in each time period $j$.**   Under this prior, the probability of infection is given by $\Phi_j$ as in Eqs 1 and 5. The infection generating process is:

$$z_{i,j} \quad \sim \quad \text{Bernoulli}(\phi_j) \tag{6}$$

$$\phi_j \quad \sim \quad h(j) \tag{7}$$

where $h$ is a user specified function describing the prior distribution on $\boldsymbol{\Phi}$, $P(\Phi_j)$. By default, $h$ is the uniform distribution, $\phi_j \sim unif(0, 1)$, though it may be set to incorporate information related to transmission such as seasonality or changes in social behaviour.

This prior is appropriate when $\boldsymbol{\Phi}$ is itself of interest and when a distinct user-specified prior, $P(\Phi_j)$, is desired for each possible infection $j$. Markov chain Monte Carlo (MCMC) mixing under this version is relatively slow given the correlation of each $\Phi_j$ and $\sum_{i=1}^{n} Z_{i,j}$, and this prior is therefore most suited when $P(\Phi_j)$ is well informed and the number of potential infection periods is small.

**Prior 2, beta prior on the probability of infection in each time period $j$.**   As in prior 1, this prior assumes that individuals are under a common infection process during a given window of time. The infection generating process is:

$$z_{i,j} \quad \sim \quad \text{Bernoulli}(\phi_j) \tag{8}$$

$$\phi_j \quad \sim \quad \text{Beta}(\alpha, \beta) \tag{9}$$

By choosing a beta prior on $\Phi_j$ with parameters $\alpha$ and $\beta$, using the analytical solution for the integral over values of $\Phi_j$ allows the infection history likelihood to be simplified:

$$P(\mathbf{Z}) \quad = \quad \prod_{j=1}^{m} \int_{0}^{1} \left( \prod_{i=1}^{n} P(Z_{i,j}|\Phi_j = \phi_j) \right) P(\Phi_j = \phi_j) d\phi_j \tag{10}$$

$$= \quad \prod_{j=1}^{m} \int_{0}^{1} \phi_j^{k_j} (1 - \phi_j)^{(n_j - k_j)} P(\Phi_j = \phi_j) d\phi_j \tag{11}$$

$$= \quad \prod_{j=1}^{m} \frac{B(k_j + \alpha, \beta + n_j - k_j)}{B(\alpha, \beta)} \tag{12}$$

where $B$ is the beta function; $k_j = \sum_{i=1}^{n_j} Z_{i,j}$ is the total number of infections during time period $j$; and $n_j$ is the number of individuals that could be infected during time period $j$. Eq 1 may then be changed to:

$$P(\mathbf{Z}, \mathbf{\Phi}, \mathbf{\Theta}|\mathbf{Y}) \propto \prod_{i=1}^{n} \left( \prod_{t \in t_i} f(\mathbf{Y}_{i,t}|\mathbf{Z}_i, \mathbf{\Theta}) \right) P(\mathbf{Z}) P(\mathbf{\Theta}) \tag{13}$$

Under this prior, the prior on the per-capita attack rate is beta distributed and the prior on the lifetime number of infections for any individual follows a binomial distribution. This version is suitable when individuals are assumed to be under the same infection generating process (e.g. in the same location) as in prior 1, but where faster MCMC mixing and convergence is required. For example, when the number of potential infection times is large, this prior significantly improves mixing by integrating out each $\Phi_j$. We have also found that this prior gives unbiased attack rate estimates when titre data are sparse and the number of individuals is large.

**Prior 3, beta-binomial prior on the total number of infections during an individual's life.** Unlike priors 1 and 2, this prior assumes that an individual's risk of infection at a given time is independent of all other individuals. Rather, a prior is placed on the total number of infections that an individual is expected to experience over the course of their life. This is the prior used in our previous work [27]. The infection generating process is assumed to be:

$$z_{i,j} \quad \sim \quad \text{Bernoulli}(\lambda_i) \tag{14}$$

$$\lambda_i \quad \sim \quad \text{Beta}(\alpha, \beta) \tag{15}$$

Similar to prior version 2, the infection history likelihood may be simplified by integrating over $\Lambda_i$:

$$P(\mathbf{Z}) \quad = \quad \prod_{i=1}^{n} P(\mathbf{Z}_i) \tag{16}$$

$$= \quad \prod_{i=1}^{n} \int_{0}^{1} P(\mathbf{Z}_i | \Lambda_i = \lambda_i) P(\Lambda_i = \lambda_i) d\lambda_i \tag{17}$$

$$= \quad \prod_{i=1}^{n} \int_{0}^{1} \lambda_i^{k_i} (1 - \lambda_i)^{m_i - k_i} P(\Lambda_i = \lambda_i) d\lambda_i \tag{18}$$

$$= \quad \prod_{i=1}^{n} \frac{B(\alpha + k_i, \beta + m_i - k_i)}{B(\alpha, \beta)} \tag{19}$$

where $B$ is the beta function; $k_i = \sum_{j=1}^{m_i} Z_{i,j}$ is the total number of infections experienced by individual $i$; and $m_i$ is the number of time periods that individual $i$ could be infected in. The posterior distribution can then be written as in Eq 13.

The prior on the per-capita attack rate across all individuals therefore follows a binomial distribution, and the prior on the lifetime number of infections for any individual follows a beta-binomial distribution, with parameters $\alpha$ and $\beta$ that can be set by the user. This prior is suitable when individuals can be assumed to be under different infection generating processes but still share antibody kinetics parameters. This version can give the quickest convergence and most efficient chain mixing when there is a relatively small number of individuals with a large amount of antibody titre data.

**Prior 4, beta prior on the probability of any infection.** In the final prior version, infection states are assumed to be independently and identically distributed with respect to both time and individual under the following infection generating process:

$$z_{i,j} \quad \sim \quad \text{Bernoulli}(\phi) \tag{20}$$

$$\phi \quad \sim \quad \text{Beta}(\alpha, \beta) \tag{21}$$

and the marginal likelihood of $Z$ is:

$$P(\mathbf{Z}) \quad = \quad \int_{0}^{1} \left( \prod_{j=1}^{m} \prod_{i=1}^{n} P(Z_{i,j} | \Phi = \phi) \right) P(\Phi = \phi) d\phi \tag{22}$$

$$= \quad \frac{B(k + \alpha, \beta + nm - k)}{B(\alpha, \beta)} \tag{23}$$

where $B$ is the beta function; $k$ is the total number of infections across all years and individuals; and $nm$ is the total number of possible infection events.

This assumption places a beta-binomial prior on both the number of infections at a given time $j$ (the attack rate) and the number of lifetime infections experienced by individual $i$. This prior is suitable when weakly informative priors are desired on both attack rates and total life-time infections per individual, and where there are a small number of individuals and small amount of titre data in the sample.

**Posterior sampling.**    We implemented a custom, adaptive MCMC framework in *serosolver* to sample from the joint posterior distribution of Θ and *Z* conditional on the antibody titre data *Y* (Eq 1). The package jointly estimates Θ and *Z* using a Metropolis-Hastings algorithm, alternating between sampling values for Θ and *Z*. The MCMC framework automatically tunes the proposal step size for Θ, and changes the number of individuals sampled for *Z* to achieve a specified acceptance rate. Because MCMC sampling of binary variables is a challenging problem in large dimensions [42, 43], *serosolver* includes custom proposal steps for *Z* to improve chain mixing. The full sampling algorithm for *Z* is described in S1 Text. Briefly, the algorithm uses a random-scan Metropolis-within-Gibbs proposal on infection histories to either propose new infection states or swap the times of existing infection states. These steps were developed to improve MCMC mixing when the infection states in adjacent time periods may be highly correlated. In the event that automated tuning is insufficient to achieve good mixing, all of the parameters controlling the proposal algorithm are exposed to the user to be changed manually from their default values.

**MCMC diagnostics.**    To ensure reliable MCMC model fitting, thorough convergence diagnostics should be calculated to ensure that separate MCMC chains have converged on the same distribution, are not trapped in local modes and provide estimates of the posterior distribution with sufficient sample size. Functions to test these criteria fall into two broad categories: (i) visual assessment of convergence and goodness of fit; (ii) metrics of convergence checking between-chain agreement, auto-correlation and effective sample size. Alongside existing tools in the *coda* and *bayestools* packages [44, 45], these functions include: MCMC trace and density plots for antibody kinetics parameters; MCMC trace and density plots for inferred attack rates over time; MCMC trace and density plots for inferred infection histories; model predicted titres plotted against observed titres; and inferred attack rates over time. MCMC chain outputs are written to disk during the fitting procedure, and the chain outputs are compatible with the *coda* and *bayesplot* R packages. The full posterior distribution of infection states as augmented data is therefore easily recoverable for further analysis, for example, for regression analysis of numbers of infections during some period of time.

## Implementation

In *serosolver*, model inputs and assumptions may be changed depending on the serological data and hypotheses under consideration. For example, in some cases the user may be most interested in short-term, fine-scale (e.g. weekly or monthly) dynamics of infection; in other situations, long-term annual dynamics may be of interest. Furthermore, although much of the development of this package came from analysis of influenza A/H3N2 dynamics, these concepts and inputs are easily adaptable to antigenically stable pathogens by specifying a null antigenic map.

The package work flow is divided into a number of distinct stages, which handle the data and parameter inputs, simulation, inference, posterior diagnostics, and analysis (Fig 2). We developed the package to rely on only a few function calls for each of these stages, but with ample room for customisation and flexibility at each stage.

To set up the model, users only need to provide: a data frame describing the model parameters (they can also change a flag to fix or estimate any of the parameters); a data frame with the antibody titre data in long format; and (optionally) an antigenic map describing the antigenic relationship between each strain. A null argument may be specified for the antigenic map when modelling only a single circulating antigenic variant. Examples of a typical data cleaning workflow are provided in S4 Text.

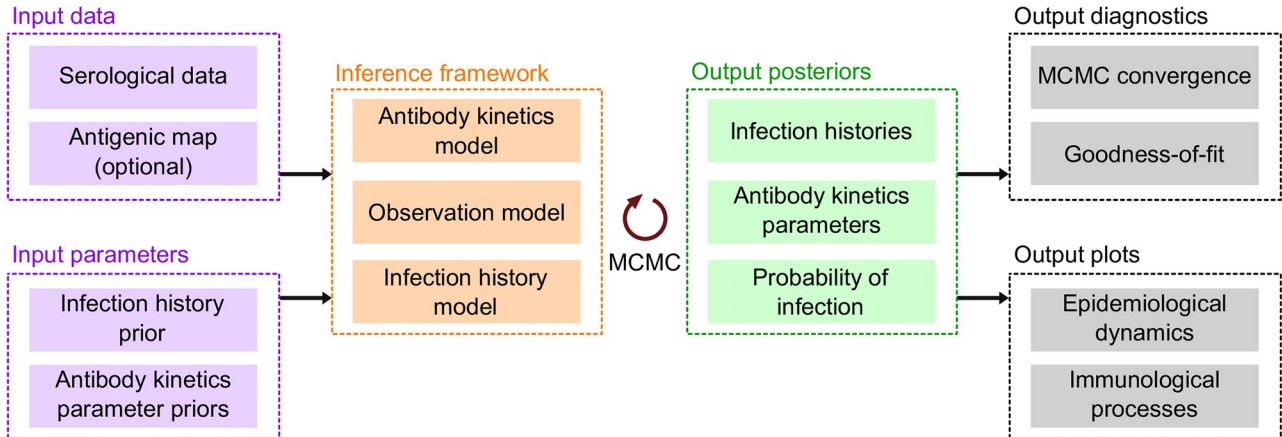

**Fig 2. Inputs and outputs for the *serosolver* R package.** Users input the serological data to be fitted, an antigenic map if considering an antigenically variable pathogen, the infection history prior and any priors on the antibody kinetics parameters. These inputs feed into the process model that can either be used to simulate data by itself, or combined with observed data and MCMC to obtain three posterior outputs: individual-level infection histories, population probabilities of infection, and antibody kinetics parameters. Once these posteriors have been obtained, *serosolver* can run MCMC diagnostics and plot key immunological and epidemiological processes.

Users may create their own likelihood and prior functions on top of those provided by default, requiring only that they return a vector of likelihoods (one per individual), and accept arguments for a vector of parameters (matching those defined in the general *serosolver* model) and the infection history matrix. Users can specify which prior assumption about infection histories is used, as specified above. In addition to the range of inbuilt options, the modular workflow of *serosolver* means that custom extensions tailored to particular problems should be readily achievable with only minor modifications to the code. In particular, alternative antibody kinetics models that capture pathogen-specific immunology and alternative assumptions about the infection history generating process.

Because of the complex nature of these Markov spaces, it is usually important to run multiple chains when using the Metropolis-Hastings MCMC algorithms implemented by *serosolver*. Multiple chains from different starting locations in the space help to detect convergence issues arising, for example, from multi-modal posterior distributions. Furthermore, model comparison and sensitivity analyses are a common output of model fitting analyses. It is simple to use *serosolver* with a parallel back-end, either through a computing cluster or locally with packages such as *doParallel* [46]. The accompanying vignettes (S3 and S4 Text) demonstrate how multiple chains may be run in parallel locally, but we note that much of our own work with *serosolver* is done using a computing cluster.

## Results

### Infection history priors

Fig 3 compares infection history metrics simulated from models under the different infection history priors compared to the analytical probability mass functions for the total number of infections per unit time $j$ and per lifetime of individual $i$. These results highlight the contribution of different prior assumptions on inferred attack rates, total number of lifetime infections and accumulation of an infection history with age. The ability of each of the different priors to recover known antibody kinetics parameter values, infection histories and attack rates is demonstrated in S1 Text using simulation-recovery experiments with different sero-survey designs

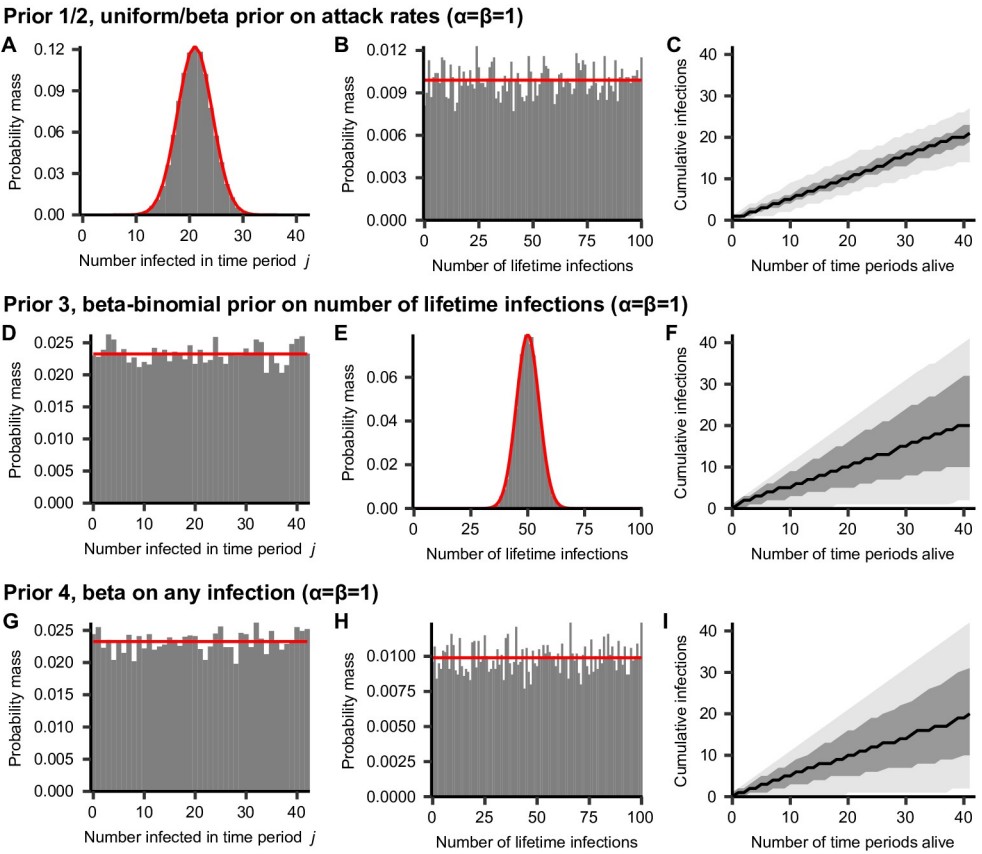

**Fig 3. Simulated vs. analytical infection history prior metrics ($\alpha = \beta = 1$).** Bars show density histograms of infections from 10,000 simulated infection histories for 100 individuals across 42 infection periods. Red lines show known probability mass function. Plots A, D and G show the prior on the total number of infections per discrete time period $j$. Plots B, E and H show prior on the total number of lifetime infections per individual. Plots C, F and I show the prior on the cumulative number of infections across 42 time periods for one individual. Black line shows prior median, dark gray region shows 50% credible intervals and light gray region shows 95% credible intervals. Note that priors 1 and 2 are equivalent under these assumptions.

under various infection history prior assumptions. Datasets may be rich in different dimensions (e.g. number of individuals vs. number of viruses tested), which leads to different inferential power for different quantities. S1 Table summarises each of these priors and the situations when each one is advised.

The different shapes and widths of the distributions in Fig 3 demonstrates how different prior forms may lead to strong assumptions about an individual's infection history. Here, a beta prior was used with $\alpha = \beta = 1$ in all of these examples (equivalent to a uniform distribution), though priors of different shapes may be set by specifying different values for $\alpha$ and $\beta$ as shown in S1 Fig. Under prior versions 1 and 2, this assumption leads to a highly constrained binomial prior on the total number of lifetime infections with mean $m\alpha/(\alpha + \beta)$ (Fig 3A), but a uniform prior on the attack rate in each time period $j$ (Fig 3B). Conversely, under prior version 3, the prior on the total number of lifetime infections follows the beta-binomial distribution, which is uniform under these values, whereas the attack rate prior follows the binomial distribution with mean $n\alpha/(\alpha + \beta)$ (Fig 3D&3E). Both the total number of infections per unit time and total number of lifetime infections are uniformly distributed under prior version 4 (Fig 3G&3H).

**Table 1. Comparison of run time and posterior sampling efficiency across a range of serosurvey designs.**

| Case study | Number of individuals | Mean run-time (minutes) | Number of observed titres | Number of time periods | $\theta$ ESS | Z ESS | $\theta$ ESS per minute | Z ESS per minute |
|---|---|---|---|---|---|---|---|---|
| 1 | 100 | 6.41 | 400 | 4 | 1020 | 5030 | 159 | 784 |
| 1 | 100 | 6.88 | 400 | 8 | 998 | 41000 | 145 | 5960 |
| 1 | 100 | 7.87 | 400 | 16 | 871 | 4650 | 111 | 591 |
| 1 | 500 | 13.8 | 2000 | 4 | 1020 | 545000 | 73.9 | 39500 |
| 1 | 500 | 14.8 | 2000 | 8 | 1000 | 25800 | 67.7 | 1740 |
| 1 | 500 | 16.8 | 2000 | 16 | 849 | 3840 | 50.5 | 229 |
| 1 | 1000 | 22.7 | 4000 | 4 | 981 | 1040000 | 43.2 | 45700 |
| 1 | 1000 | 26.9 | 4000 | 8 | 987 | 22000 | 36.7 | 817 |
| 1 | 1000 | 31.4 | 4000 | 16 | 913 | 4540 | 29.1 | 145 |
| 2 | 100 | 12.2 | 800 | 41 | 2030 | 2990 | 167 | 245 |
| 2 | 100 | 18.8 | 4100 | 41 | 1360 | 1980 | 72.4 | 105 |
| 2 | 100 | 19 | 800 | 82 | 1190 | 2400 | 62.4 | 126 |
| 2 | 100 | 34.2 | 8200 | 82 | 1070 | 1660 | 31.3 | 48.6 |
| 2 | 100 | 37 | 800 | 164 | 1980 | 2470 | 53.5 | 66.9 |
| 2 | 500 | 38 | 4000 | 41 | 1730 | 1860 | 45.4 | 48.9 |
| 2 | 500 | 51.2 | 4000 | 82 | 1630 | 2500 | 31.8 | 48.8 |
| 2 | 500 | 72.3 | 20500 | 41 | 1600 | 651 | 22.1 | 9.01 |
| 2 | 1000 | 73.6 | 8000 | 41 | 1580 | 910 | 21.4 | 12.4 |
| 2 | 500 | 78.6 | 4000 | 164 | 1550 | 2420 | 19.7 | 30.8 |
| 2 | 100 | 87.8 | 16400 | 164 | 846 | 2100 | 9.63 | 23.9 |
| 2 | 1000 | 90.4 | 8000 | 82 | 1550 | 2050 | 17.2 | 22.7 |
| 2 | 500 | 150 | 41000 | 82 | 925 | 478 | 6.17 | 3.19 |
| 2 | 1000 | 153 | 41000 | 41 | 1530 | 555 | 9.99 | 3.63 |
| 2 | 1000 | 182 | 8000 | 164 | 1250 | 2270 | 6.89 | 12.5 |
| 2 | 1000 | 327 | 82000 | 82 | 926 | 213 | 2.83 | 0.65 |
| 2 | 500 | 346 | 82000 | 164 | 553 | 837 | 1.6 | 2.42 |
| 2 | 1000 | 674 | 164000 | 164 | 310 | 416 | 0.46 | 0.618 |

## Computational performance

The *serosolver* code uses a C++ back-end with substantial optimisation to scale the model to large datasets and high infection time resolutions with reasonable run times. Table 1 displays the mean run time of 5 MCMC chains fitting the *serosolver* model to serological data of different dimensions. In the most complex scenario, which involves fitting the model to 164,000 antibody titre measurements and inferring the infection state of 1000 individuals at 164 different time points (164,000 infection states), effective sample sizes >200 are achievable for both the antibody kinetics parameters and attack rate estimates in <12 hours. For smaller scale analysis (e.g. 100 individuals, <5000 titres), high effective sample sizes and well-mixed chains are easily generated within 30 minutes.

## Case study results

We present two case studies to highlight the range of insights that *serosolver* can generate from serological samples. These cover two types of study designs commonly used to examine epidemiological and immunological dynamics using serological data, which can be thought of as subsets of the observations shown in Fig 1, bottom panel. The first is a serological survey testing individuals against a single homologous strain, which can reveal short-term epidemic

dynamics, analogous to observing each of the bars of a single colour from Fig 1. We use data from a longitudinal study conducted in Hong Kong between 2009 and 2011 to estimate short-term antibody kinetics parameters against A/H1N1pdm09 in a population with limited prior immunity [47]. The second type of study design involves testing samples against a panel of previously circulating strains, which can provide insights into historical patterns of infection, analogous to observing all of the bars within a single serum sample from Fig 1 [37, 48]. To illustrate this application, we apply the package to cross-sectional samples tested against a panel of historical A/H3N2 influenza strains to infer infection histories and antibody kinetics [37, 48].

**Case study 1.** The first case study uses data from a cohort study in Hong Kong during and after the 2009 A/H1N1pdm09 outbreak [47]. With repeat serological samples tested against a given virus, *serosolver* can reconstruct the unobserved infection dynamics from measured titres collected several months apart. It is also possible to examine these infection dynamics stratified by available demographic variables, such as vaccination status (Fig 4A) and age (Fig 4B). Fig 4C shows example model fits to the observed antibody titres. Finally, we can estimate biological parameters shaping the short-term antibody response (Fig 4D).

We estimated quarterly exposure rates, which could include either infection or vaccination. The inferred peaks in exposure rates are consistent with the observed two waves of the 2009 pandemic [47]. Constrained estimates of high incidence were obtained for Q4 2009. The following period of elevated incidence but with high uncertainty in early 2010 reflects the gap between the second and third round of serum sampling; further antibody boosts were detected in this period, but the exact quarter could not be determined.

We investigated the impact of vaccination status and age on inferred exposure rates, finding differences in exposure rates in vaccinated individuals (n = 113) compared to unvaccinated individuals (n = 307). Inferred boosting rates were almost identical between vaccinated and unvaccinated individuals up to and including Q4 2009. However, higher overall exposure rates in vaccinated individuals were inferred from Q1 2010 onward (Fig 4A). Additionally, we observed clear differences in age-stratified exposure rates of unvaccinated individuals (Fig 4B), with exposure rates highest among children (<19 years old, n = 30) and adults (19-64 years old, n = 264), and lowest among the elderly (>64 years old, n = 17), confirming previous findings of age-stratified exposure rates during the 2009 pandemic [49]. Some of the inferred infections in Q1 2009 may represent pre-existing baseline titres rather than infections with A/H1N1/pdm09.

The pandemic vaccine was available from December 2009, and this increase in inferred boosts from Q1 2010 may therefore capture pandemic vaccine-derived responses. Intuitively, we would expect infection rates to be lower in vaccinated individuals; however, the converse suggests that vaccination caused additional antibody boosting. It is unlikely that these detected boosts represent true infections, as the vaccinated cohort had a larger proportion of older individuals, who likely experienced lower infection incidence (Fig 4B). Vaccination around the first sampling round was predominantly with the older 2009/10 seasonal vaccination, which was unlikely to elicit a novel antibody response (rather than a memory-derived response) effective against the antigenically dissimilar pandemic virus.

We aimed to characterise the short-term immune response following infection by estimating long and short-term antibody kinetics parameters. We found that there is a strong short-term average boost ($\mu_s$) of 2.59 (posterior median; 95% CI: 2.19-2.86) log titre units followed by a persistent long-term boost ($\mu_l$) of 3.38 (posterior median; 95% CI: 3.27-3.52) log titre units. We estimated that every 3 months, 58.4% (posterior median; 95% CI: 48.5-72.6%) of the short-term boost is lost ($\omega$) such that only the long-term boost remains after 0.428 years (posterior median; 95% CI: 0.344-0.515).

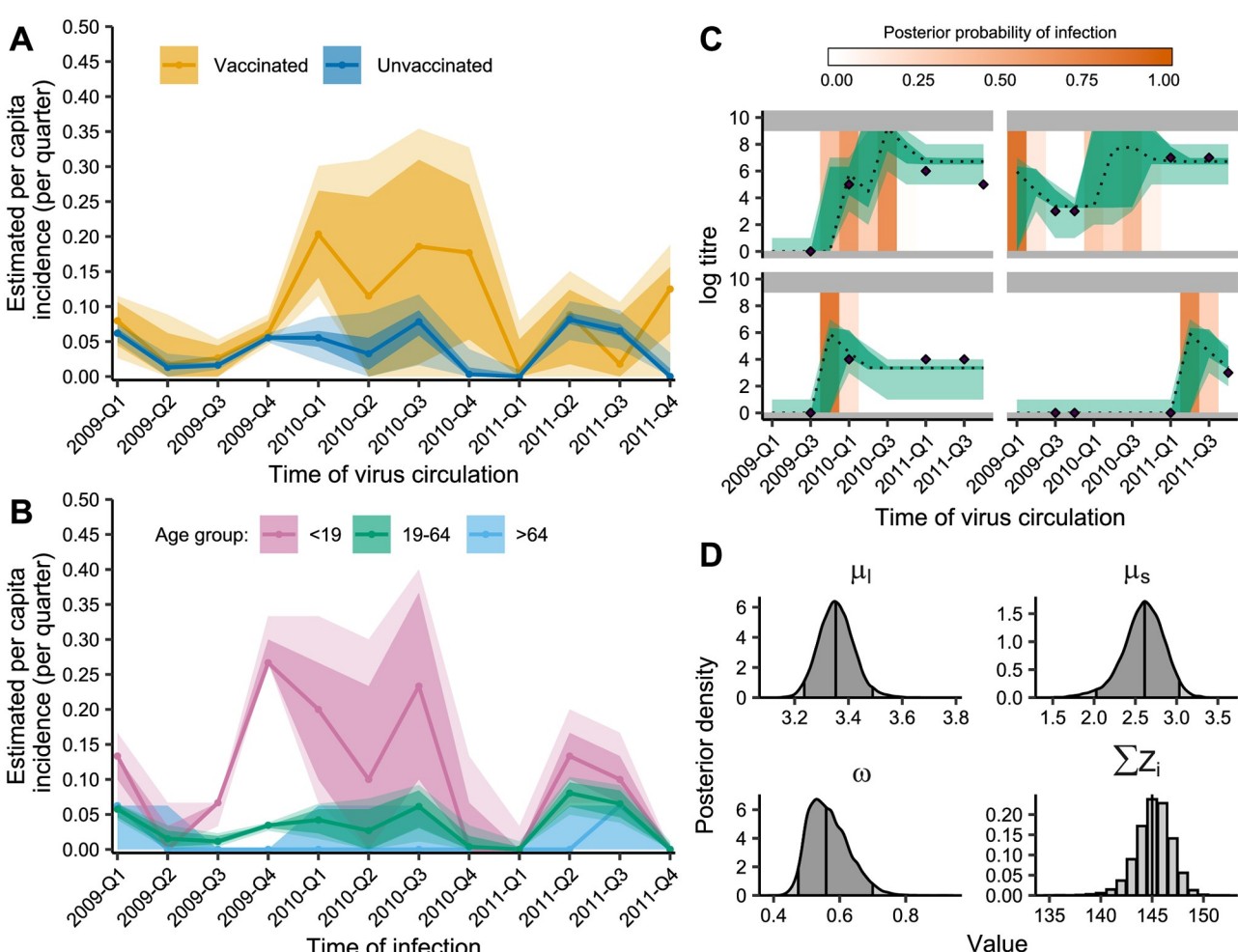

**Fig 4. Influenza A/H1N1pdm09 infection dynamics in Hong Kong cohort.** A: Exposure rates in unvaccinated and vaccinated individuals. Shaded regions show 80% (dark) and 95% (light) credible intervals (CI). Solid lines shows posterior medians. X-axis gives midpoint for that quarter. B: Age-specific exposure rates in unvaccinated individuals. Solid lines show median estimates for each age group (pink: <19 (n = 30), green: 19-64 (n = 264), blue: >64 (n = 17)) with 80% (dark) and 95% (light) CI shaded. C: Model predicted titres and inferred infections compared to observed titres for 4 representative individuals with inferred infections. Purple diamonds show observed titres; black dashed lines indicate posterior median model predicted titres; green shading shows 95% CI on model predicted latent titres (dark) and assay observations (light); orange shading indicates posterior probability of infection. Grey region shows titres outside the limit of detection. X-axis gives midpoint for that quarter. D: Posterior densities of antibody kinetics parameters and total number of infections ($\sum Z_i$). Vertical lines represent 2.5th, 50th, and 97.5th percentiles.

To assess whether data contain enough information to reliably estimate the infection histories and biological process parameters, *serosolver* can be used to run a simulation recovery study. For example, if data of the same structure as the A/H1N1pdm09 outbreak in Hong Kong are generated using plausible parameter values [27], it is possible to re-infer these parameters (Fig 5B) alongside the individual-level infection histories (Fig 5C) and overall probabilities of infection (Fig 5A). However, depending on the sampling frequency, number of tested strains and number of repeat measurements, there are varying levels of information to estimate these quantities. When antibody titre data is sparse, the priors placed on either the antibody parameters, infection histories or probability of infection parameters will have a greater effect on the estimation performance. We therefore recommend routine implementation of

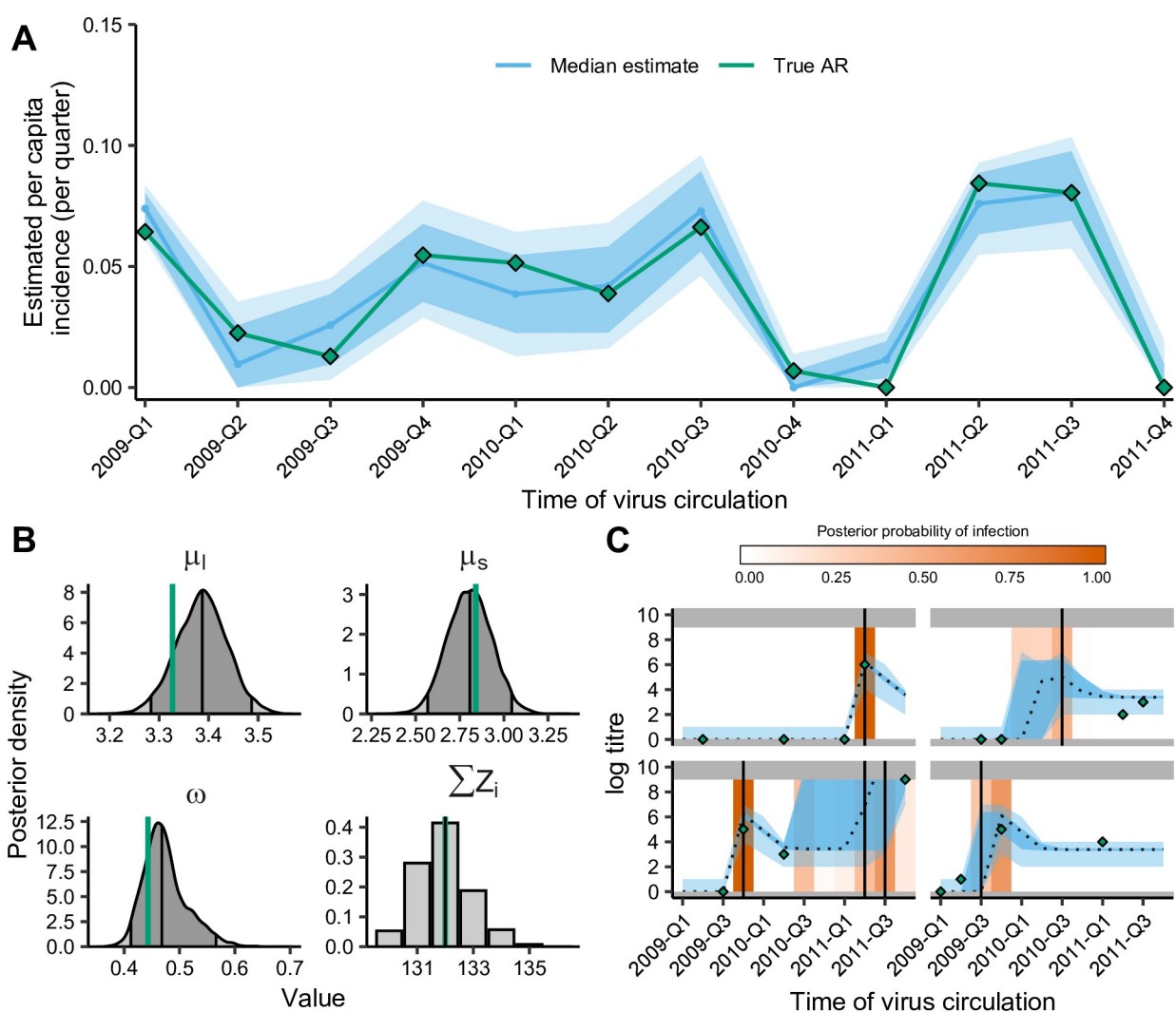

**Fig 5. Simulation-recovery of parameter and infection estimates using simulated single strain longitudinal data in same format as the Hong Kong dataset.** A: Model estimated attack rates vs. 'true' attack rates. Solid line shows estimated attack rate with 80% (dark) and 95% (light) credible intervals (CI); green line and points shows true attack rates. B: 'True' process parameters used for simulation compared to estimated posterior densities. Green vertical lines indicate true parameter values; vertical lines represent 2.5th, 50th, and 97.5th percentiles. C: Model predicted titres and inferred infections compared to observed titres and known infections. Green diamonds indicate observed titres; black dashed lines indicate posterior median model predicted titres; blue shading shows 95% CI on model predicted latent titres (dark) and assay observations (light); vertical lines indicate the timings of true infections; orange shading indicates posterior probability of infection.

simulation recovery on new data to ensure that the most suitable model is being applied to the data available.

**Case study 2.** The second case study considers cross-sectional serological samples collected in southern China in 2009, which were tested against nine historical influenza A/H3N2 strains that circulated between 1968 and 2008 [37, 48]. We demonstrate how *serosolver* can be used to reconstruct several features of the epidemiological and immunological dynamics in this cohort. First, Fig 6A shows substantial variation in the inferred historical attack rates of A/H3N2, with clear periods of high incidence interspersed by periods of very low incidence (range of posterior medians: 3.63% to 95.2%). Periods of high and low attack rates were similar to those in a previous analysis from a cohort in Ha Nam, Vietnam [27]. In these analysis, we

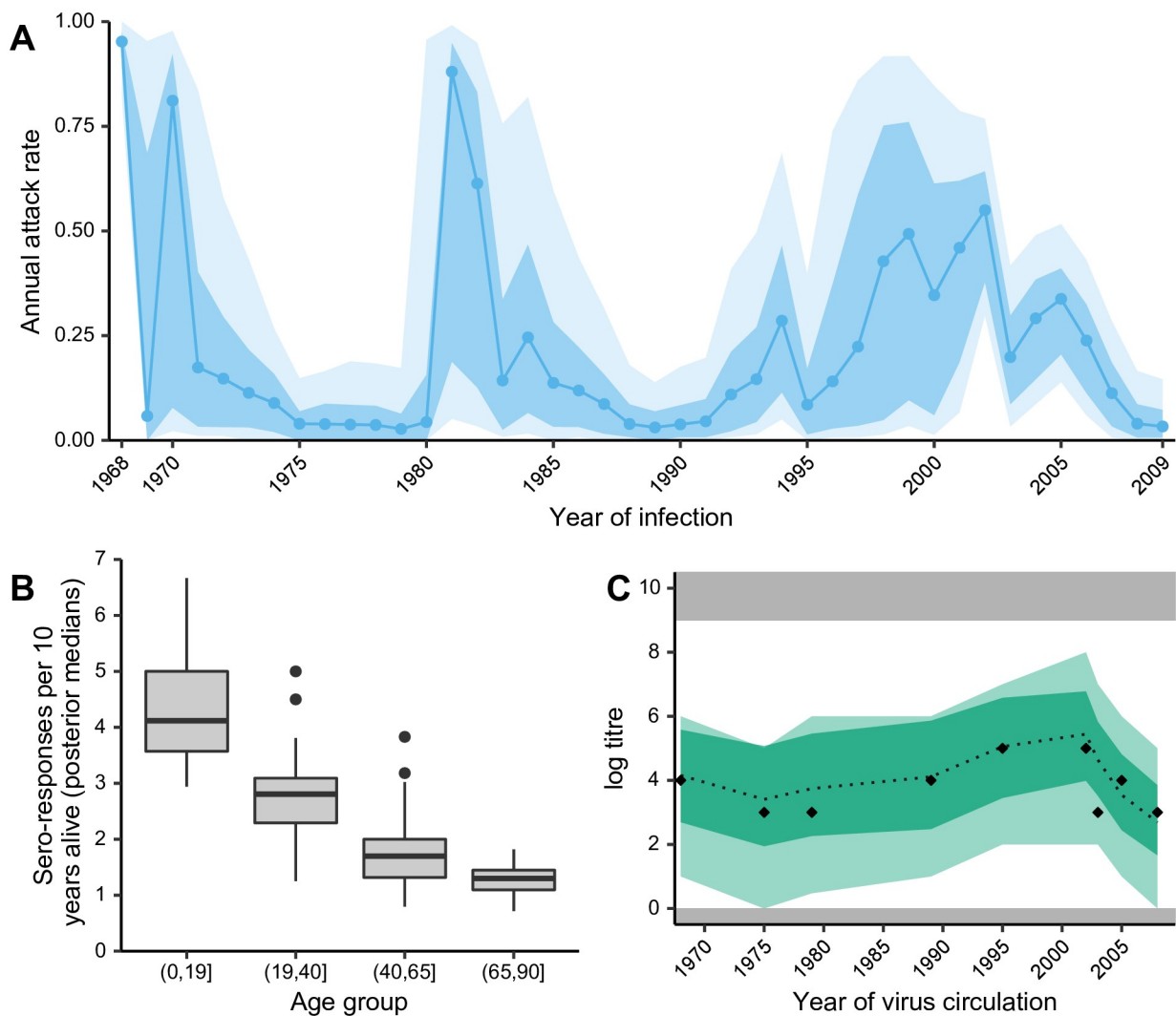

**Fig 6. Influenza A/H3N2 dynamics in southern China.** A: Inferred historical attack rates. Shaded regions show 80% and 95% credible intervals (CI), solid line and points shows posterior median estimate; B: Frequency of inferred antibody responses (sero-responses) by age group. Boxplots show distribution across individuals based on posterior median total number of infections per individual per 10 years alive. C: Model predicted titres and inferred infections compared to observed titres (black diamonds). Shaded regions show 95% CI on model predicted latent titres (dark) and assay observations (light).

used a weakly informative prior on the annual attack rate with a mode of 15% with prior version 2. Our posterior estimates were very similar to this, with a median inferred attack rate of 14.6%, suggesting either agreement between the data and prior or a lack of information in the data.

We also identified clear age-specific patterns of infection. Fig 6B shows the median number of infections per 10 years alive stratified by age at the time of exposure. These estimates agree with previous analyses that individuals are infected, or at least experience antibody boosting, less frequently as they get older [27]. Inference of long-term biological parameters suggested that individuals experience a long-term average antibody boost $\mu_l$ of 2.24 log units (posterior median; 95% CI: 1.95-2.51), corresponding to approximately a 4-fold boost to long-term homologous titres that wanes with antigenic distance (long-term cross reaction $\sigma_l = 0.105$

posterior median; 95% CI: 0.0962-0.113) and decreases with each successive exposure (antigenic seniority parameter, $\tau$ = 0.0310 posterior median; 95% CI: 0.0210-0.0415).

As with the first case study, simulation recovery was used to validate the ability of *serosolver* to correctly infer underlying processes from a given dataset (discussed in detail in S4 Text).

## Availability and future directions

The *serosolver* package provides a general inference framework to estimate epidemiological and immunological dynamics from serological data. The open source package is available from GitHub (https://github.com/seroanalytics/serosolver), with detailed accompanying vignettes covering the main implementation and case studies presented here. The aim of this package is to provide an open source, modifiable code base to fit antibody kinetics models that also require inference of unobserved infections. Disparate serosurveys measuring antibody titres over time are often underpinned by comparable dynamics, and we therefore felt that a unifying tool to enable quick reproduction and direct comparison of analyses across different datasets would be a useful addition to the literature.

As well as the stand-alone applications we have illustrated in the case studies above, *serosolver* could easily link with traditional epidemiological analysis. The results presented here are not intended to be exhaustive analyses, but rather demonstrate the utility and range of insights that can be generated from serological data. In particular, the posterior latent individual-level infection histories and titre trajectories could act as observations for regression models. For example, *serosolver* outputs could be combined with syndromic or lab-confirmation data to examine the relationship between susceptibility and titre at time of infection [50]. These methods could also apply to other pathogens; a similar model structure has recently been used to examine latent titres for dengue [35]. Although our work so far has focused on influenza and therefore uses an antigenic map to specify cross-reactivity between related influenza strains, alternative models to describe cross-reactivity in serological assays could be coded into *serosolver* as part of the antibody kinetics model. For example, independent parameters for the pairwise cross-reactivity of each pathogen in the system could be inferred directly given sufficient serological data.

The simulation-recovery results presented here highlight that different priors have their uses depending on the distribution of the serological data, the resolution of the model and the particular question under consideration. If the data being fitted has few individuals, few infection times and a large number of titres, then the assumptions of these priors has relatively little impact on the inferred infection histories. However, if the amount of data and therefore weighting of the likelihood is small, then the infection history prior becomes important. For example, a dataset with very few individuals but a large number of tested titres per individual may be well suited to analysis under the beta-binomial prior on total lifetime infections (prior 3) where the aim is to infer an individual's lifetime infection history and antibody kinetics parameters, but not necessarily population-level attack rates. Conversely, inferring accurate historical attack rates is better suited to the priors on per time attack rates (priors 1 and 2), as infection probabilities are shared across individuals.

Prior knowledge on the time of exposure may be incorporated into *serosolver*, either from surveillance data or, if relevant, temporal climate variables. In the case studies presented, we used relatively simple priors for the probability of infection. However, more complex temporal priors could be imposed by having a different prior distribution for the probability of at each time point (i.e. different values of $\alpha$ and $\beta$) to account for seasonality in transmission dynamics. In the future, we hope to extend *serosolver* to include non-linear feedback between past exposures and future risk by embedding an epidemic model as well as the probability of

infection [32]. In theory, this package could be used to generate an ongoing database of inferred immunological parameters, allowing estimates to be updated and combined across studies to better estimate attack rates and infection histories in less data-rich cohorts.

This framework could also be used to inform the design of serological sample collection and testing. Given potential logistical or budgetary restrictions on analysis of stored sera or collection of new samples, *serosolver* could be used to simulate different study designs and show how accurately these designs could recover the main parameters of interest.

At present, *serosolver* focuses on inference for a single exposure type. However, for viruses like influenza and dengue, individuals may be exposed to multiple subtypes or serotypes in the same season. Exposure to one antigen may cross react with another antigen providing protection against antigens an individual has not been directly exposed to. For example, infection with influenza A/H1N1 may provide cross-reactive protection against other group 1 viruses, and A/H3N2 against group 2 viruses [51]. Additionally, the incorporation of multiple exposures can facilitate the inclusion of vaccine exposure. In influenza, where vaccination is recommended annually, exposure to vaccination is an important piece of the immunological life course puzzle of an individual [52]. In its current form, *serosolver* can estimate differences between exposures by being fit independently to different subtypes. It can also fit models separately to vaccinated or unvaccinated populations to estimate how serological dynamics vary between these groups. Although this is a useful first approximation, future versions of *serosolver* will include potential for multiple exposure types during the same season so that any interactions can be modelled explicitly. Such an extension will also allow for multiple co-circulating pathogens to be modelled.

Through using a multi-strain HI assay panel, we estimated historical influenza A/H3N2 attack rates in southern China from 1968 onward. The assumption that the probability of infection was the same for all individuals in the cohort at a given time faithfully captures the expected uncertainty in estimates from times when only few individuals in the cohort were alive. Further data from the Fluscape study with increased strain coverage, number of individuals, and repeated longitudinal serum samples will help improve the precision of these estimates [53]. However, attack rate estimates of nearly 100% must be interpreted with caution. In the model, elevated titres to historical strains may only be explained by a fixed amount of homologous and cross-reactive antibody boosting from infection. There are other ways in which relatively high titres might be achieved: different strains may elicit different levels of homologous boosting [18]; systematic bias in the HI assay towards higher titres in particular strains may falsely suggest higher exposure rates [26, 39]; survival of high-immunity individuals may bias the sample; and continual strengthening of titres from asymptomatic exposures should be considered. Alternatively, these estimates may be a true reflection of infection, or at least antibody response, rates that are far higher than previously thought based on traditional methods. Epitope-specific antibody kinetics models combined with state-of-the-art assays may be useful in disentangling the relative contributions of these effects [54, 55].

There is increasing evidence that serological titre data contain substantial additional information about infection and immunity dynamics, which are not captured by simple 4-fold rise metrics [16, 28, 29, 35] Furthermore, in multi-strain pathogen systems, evidence is mounting that individual-level heterogeneity in unobserved exposure histories is a key driver of susceptibility to infection and disease [26, 52, 54, 56]. The *serosolver* package provides a generic framework to extract this information from commonly collected data. As serological data become increasingly available, it will be important to develop modern analytical methods and tools that account for known biological and epidemiological processes that may confound or bias inference [29, 57–59].

## Supporting information

**S1 Text. General statistical framework and derivation of the infection history priors.**
(PDF)

**S2 Text. Additional antibody kinetics and code update guide.**
(PDF)

**S3 Text. Case study 1 vignette with all code required for model fitting, figure generation and simulation recovery.**
(HTML)

**S4 Text. Case study 2 vignette with all code required for model fitting, figure generation and simulation recovery.**
(HTML)

**S1 Table. Summary of infection history priors.**
(PDF)

**S1 Fig. Simulated vs. analytical infection history prior metrics ($\alpha = 2, \beta = 10$).** Bars show density histograms of infections from 10,000 simulated infection histories for 100 individuals across 42 infection periods. Red lines show known probability mass function. Plots A, D and G show the prior on the total number of infections per discrete time period $j$. Plots B, E and H show prior on the total number of lifetime infections per individual. Plots C, F and I show the prior on the cumulative number of infections across 42 time periods for one individual. Black line shows prior median, dark gray region shows 50% credible intervals and light gray region shows 95% credible intervals. Note that priors 1 and 2 are equivalent under these assumptions.
(PDF)

## Author Contributions

**Conceptualization:** James A. Hay, Justin Lessler, Derek A. T. Cummings, Adam J. Kucharski, Steven Riley.

**Data curation:** James A. Hay.

**Formal analysis:** James A. Hay, Amanda Minter, Kylie E. C. Ainslie, Adam J. Kucharski.

**Investigation:** James A. Hay, Amanda Minter, Kylie E. C. Ainslie, Steven Riley.

**Methodology:** James A. Hay, Justin Lessler, Adam J. Kucharski, Steven Riley.

**Software:** James A. Hay, Amanda Minter, Adam J. Kucharski.

**Supervision:** Adam J. Kucharski, Steven Riley.

**Validation:** James A. Hay, Amanda Minter, Kylie E. C. Ainslie, Adam J. Kucharski, Steven Riley.

**Visualization:** James A. Hay, Kylie E. C. Ainslie, Adam J. Kucharski.

**Writing – original draft:** James A. Hay, Amanda Minter, Kylie E. C. Ainslie, Adam J. Kucharski, Steven Riley.

**Writing – review & editing:** James A. Hay, Amanda Minter, Kylie E. C. Ainslie, Justin Lessler, Bingyi Yang, Derek A. T. Cummings, Adam J. Kucharski, Steven Riley.

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
