## [Decision Letter · Decision Letter 0]

18 Feb 2020

Dear Dr Hay,

Thank you very much for submitting your manuscript "An open source tool to infer epidemiological and immunological dynamics from serological data: serosolver" for consideration at PLOS Computational Biology. As with all papers reviewed by the journal, your manuscript was reviewed by members of the editorial board and by several independent reviewers. The reviewers appreciated the attention to an important topic. Based on the reviews, we are likely to accept this manuscript for publication, providing that you modify the manuscript according to the review recommendations.

Sincerely,

Roland R Regoes

Associate Editor

PLOS Computational Biology

Virginia Pitzer

Deputy Editor

PLOS Computational Biology

[LINK]

Reviewer's Responses to Questions

**Comments to the Authors:**

Reviewer #1: Thank you for addressing my previous comments. Very nice paper!

Very minor suggestion: I agree with the decision to use lower case consistently with the package name. However you can avoid starting sentences with 'serosolver' by slight rephrasing. For instance:

L58: serosolver takes assay results... -> The serosolver package takes assay results...

L178: serosolver tracks each individual's infection history as a... -> Each individual's infection history is tracked by serosolver as a...

Might help with the blood pressure of grammar nazi readers.

Reviewer #2: The authors did significant revisions of the manuscript in response to Reviewers comments.

All my concerns from my previous review of the manuscript have been addressed.

It is a well-written, well-organized paper describing nicely documented flexible R package serosolver that will be of interest for those aiming to dissect humoral immune responses to multi-strain pathogens like influenza. I strongly recommend it for publication.

Correction is needed for revised Fig 4B:

Lns 450-453: text refers to the Fig 4B with three age-specific exposure rates, while revised figure shows only two age groups. Corresponding figure legend refers to 3 groups but the green color for age 19-64 on the figure does not match the color indicated in the legend.

Reviewer #3: I previously reviewed this paper for PLoS Biology and suggested that PLoS Comp Bio might be the more suitable journal. The authors addressed all comments I provided on my previous review, I have no further ones.

One general comment, mainly meant for the editor: Instead of this being labeled a 'research article', it might be better to classify it as a 'software article' (or whatever terminology the journal uses).

**Have all data underlying the figures and results presented in the manuscript been provided?**

Reviewer #1: Yes

Reviewer #2: Yes

Reviewer #3: Yes

PLOS authors have the option to publish the peer review history of their article (what does this mean?). If published, this will include your full peer review and any attached files.

Reviewer #1: No

Reviewer #2: No

Reviewer #3: No
---

## [Editor Report · Decision Letter 1]

1 Apr 2020

Dear Dr Hay,

We are pleased to inform you that your manuscript 'An open source tool to infer epidemiological and immunological dynamics from serological data: serosolver' has been provisionally accepted for publication in PLOS Computational Biology.

Best regards,

Virginia E. Pitzer, Sc.D.

Deputy Editor

PLOS Computational Biology

Virginia Pitzer

Deputy Editor

PLOS Computational Biology

---

## [Editor Report · Acceptance letter]

20 Apr 2020

PCOMPBIOL-D-20-00058R1 

An open source tool to infer epidemiological and immunological dynamics from serological data: serosolver

Dear Dr Hay,

I am pleased to inform you that your manuscript has been formally accepted for publication in PLOS Computational Biology. Your manuscript is now with our production department and you will be notified of the publication date in due course.

With kind regards,

Laura Mallard
